# Israeli *Rousettus aegyptiacus* Pox Virus (IsrRAPXV) Infection in Juvenile Egyptian Fruit Bat (*Rousettus aegyptiacus*): Clinical Findings and Molecular Detection

**DOI:** 10.3390/v13030407

**Published:** 2021-03-04

**Authors:** Dan David, Irit Davidson, Sharon Karniely, Nir Edery, Ariela Rosenzweig, Asaf Sol

**Affiliations:** 1Kimron Veterinary Institute, Bet Dagan 50250, Israel; iritd@moag.gov.il (I.D.); sharonk@moag.gov.il (S.K.); nire@moag.gov.il (N.E.); asafs@moag.gov.il (A.S.); 2The Israeli Wildlife Hospital and Safari, Zoological Center, Tel Aviv 5225300, Israel; arielaros@yahoo.com

**Keywords:** Egyptian fruit bat (*Rousettus aegyptiacus*), poxvirus, IsrRAPXV, skin lesions, tongue lesions

## Abstract

During 2019, five carcasses of juvenile Egyptian fruit bats (*Rousettus aegyptiacus*) were submitted to the Kimron Veterinary Institute. These bats exhibited typical poxvirus like lesion plaques of different sizes on the skin, abdomen and the ventral side of the wings. Clinical and histopathological findings suggested a poxvirus infection. Infectious virus was isolated from skin swabs, skin tissue and tongue of the dead bats and was further confirmed to be a Poxvirus by molecular diagnosis using PCR with pan-chordopoxviruses primers. All the dead bats were found positive for two Poxvirus genes encoding a metalloproteinase and DNA dependent DNA polymerase. In this study, a novel real time quantitative PCR (qPCR) assay was established to further confirmed the presence of specific poxvirus viral DNA in all pathologically tested tissues. Moreover, according to sequence analysis, the virus was found to be highly similar to the recently discovered Israeli *Rousettus aegyptiacus* Pox Virus (IsrRAPXV).

## 1. Introduction

Bats make approximately 20% of living mammals and are extremely disperse species, found in all continents except Antarctica [1,2]. Recently there is an emerging interest in bats since they recognized to be hosts for large swath of pathogens that affect humans and livestock [3]. In fact, bats are considered the natural reservoir of a large variety of zoonotic viruses including henipaviruses [4], filoviruses [5], some lyssaviruses [6], SARS-CoV [7], Middle East Respiratory Syndrome [8] and SARS-CoV-2 [9].

To date, 33 bat species have been identified in Israel. The Egyptian fruit bat is the only megachrioptera amongthese species and the most common bat found in close proximity to humans in urban areas, national parks and nature reserves [10]. As true for other bats species, Egyptian fruit bats can harbor many different viruses without showing clinical signs and are the natural reservoir for the Marburg [5] and Sosuga viruses [11].

Poxviruses are double-stranded DNA viruses with relatively large genome (150–300 kbp). They are divided into two subfamilies, the invertebrate-infecting Entomopoxvirinae and the chordate-infecting Chordopoxvirinae [12]. Poxviruses were found to be involved in different diseases in bats [13,14] including the recent reports suggesting the association of poxvirus with a clinical disease of the skin in infected bats [15,16]. In 2016, O’Dea et al., described pox like lesions across the entire area of the wing membrane of a flying fox bat in Australia [16]. Recently we reported the identification of a previously unrecognized poxvirus-IsrRAPXV, which we found to be involved with a clinical disease of the skin of adult Egyptian fruit bats in Israel [15].

Here we describe the clinical and pathological findings of juvenile Egyptian fruit bats infected with poxvirus like -IsrRAPXV. Moreover, we describe a novel qPCR assay for detection of IsrRAPXV in target tissues.

## 2. Materials and Methods

### 2.1. Bats: Sampling and Processing

Five juvenile suckling Egyptian fruit bats (*Rousettus aegyptiacus*) were used in this study (Table 1). Three bats collected from the northern part of Israel (1334, 1374 and 1375) and 2 bats from the central region of Israel (1840, 1827). The bats were admitted to the Wildlife medical center at the Safari Zoo, Ramat Gan. All the bats were treated with antimicrobial drugs, nutrition and fluid support. Following medical inspection, skin swabs from lesions were collected from three bats (1334, 1374 and 1375). The bats showed progressive deterioration over 6–8 days (Table 1) concomitant with an increase in the severity of the pox like lesion on the ventral wings. Upon death, bats carcasses were submitted for pathological examination at the Kimron Veterinary Institute, Bet Dagan, Israel. At the necropsy, skin swabs (from bats 1827 and 1840) and organs from all bats were collected and kept at −80 °C for further investigation.

### 2.2. Virus Isolation

Skin swabs specimens were suspended in 2 mL Phosphate Buffered Saline (PBS), incubated for 1 h at room temperature and then clarified by centrifugation at 1000× *g* for 10 min at 4 °C. The supernatants were passed through a 0.45 µm membrane filter and kept at −80 °C. Organs tissues and skin samples were homogenized in 2 mL PBS, centrifuged at 1000× *g* for 10 min at 4 °C and supernatant was collected and passed through a 0.45 µL membrane filter and kept at −80 °C. Filtered samples were used for virus isolation using African green monkey kidney epithelial Vero cell cultures (CCL 81™, ATCC^®^,VA, USA) grown in 6- well plates (Corning, New York, NY, USA). Briefly, 0.5 mL of filtrates was applied on cells and incubated for 1 h at 37 °C and 5% CO_2_. Following incubation, 4 mL of maintenance medium, EMEM (Biological Industries, Bet hahemek, Israel) supplemented with 2% fetal bovine serum (Gibco, Invitrogen, CA, USA), 2% glutamine, and 1% antibiotics (Biological Industries, Bet hahemek, Israel), was added and the cells culture were inspected daily.

### 2.3. DNA Extraction and PCR Amplification

Total DNA was extracted from target tissues (~10 mg) using the DNA Easy blood and tissue kit (Cat no. 69506, Qiagene, Copenhagen, Denmark) as recommended by the manufacturer. Purified DNA was tested in a diagnostic pan-poxvirus PCR test targeting, Insulin metalloproteinase and DNA-dependent DNA polymerase genes as previously described [17,18]. Briefly, PCR of poxvirus DNA-dependent DNA polymerase and viral insulin metallotransferase like protein genes was performed using previously described primers amplifying 338 bp and 220 bp sequences, respectively ([17,18], Table 2). PCR conditions used for the reaction were as follows: initial denaturation of 5 min at 95 °C, followed by 45 cycles of denaturation at 95 °C for 1 min, annealing at 45 °C for 1 min and extension at 72 °C for 1 min, followed by a final elongation step at 72 °C for 10 min.

PCR amplicon of viral insulin metallotransferase like protein was the same as described above except the annealing temperature set to 50 °C. PCR amplification of the two fragments were performed using Dream Taq Green enzyme PCR Master Mix according to manufacturer instructions (Thermo Scientific, Carlsbad, CA, USA) and were subjected to electrophoresis on 1.5% agarose gel. PCR products were purified and submitted to sequencing (Hylabs, Rehovot, Israel). Sequences were analyzed using the Geneious platform version 9.1.5 (Biomatters Ltd., Auckland, New Zealand) and submitted to NCBI with the following accession numbers references MT911989-MT911993 for insulin metalloproteinase and MT911994—MT911998 for DNA Dependent DNA polymerase gene.

### 2.4. Quantitative Real Time PCR (qPCR)

For detection of viral DNA in bats organs and skin swabs, we designed a novel IsrRAPXV specific quantitative PCR assay. To this end, we used the RealTimeDesign qPCR Assay Design Software (Biosreach Technologies, Middlesex, UK) based on the complete gene sequence of ORF E9L (Genbank accession no. MK542650) of IsrRAPXV strain 1000/15 as described in Table 2. First, an amplicon of 432 bp primers containing the target sequence of the qPCR assay was amplified by PCR using primers 037F—037411R (Table 2). The amplified 432 bp PCR product was separated by electrophoresis and gel purified using Expin kit (GeneAll, Seoul, Korea). Purified fragment was cloned into pGEM^®^ Easy Vector System II (Promega, Madison, WI, USA) and transformed into competent *Escherichia coli*. Positive clones were propagated in LB broth supplemented with 100 µg/mL ampicillin. Plasmid was purified using a plasmid purification kit (Promega, Madison, WI, USA) and used as a positive control template. qPCR reaction which composed of 1 µL of each primers IsrRAPXV-F and IsrRAPXV-R at concentration of 500 nM, 1 µL of probe IsrRAPXV at concentration of 250 nM, 10 µL of Real Time PCR Master mix (PerfeCTa qPCR ToughMix, Quanta, Beverly, MA, USA), 5 µL of extracted DNA and water to a final volume of 20 µL. For detection of poxvirus in bats tissues, DNA was measured and equal amounts of DNA were used per reaction. This approach enable us to compare viral load in tissues in each bat tested. The PCR reaction was run on a CFX96 thermocycler (Bio-Rad, Hercules, CA, USA) with the following cycling conditions: 50 °C for 2 min, 95 °C for 2 min following by 40 cycles of 95 for 15 s and 60 °C for 45 s.

### 2.5. Histology

Skin, wings lesions and organs were obtained and preserved in 4% neutral buffered formalin. Fixed samples were embedded in paraffin blocks, sectioned and stained with Hematoxylin and Eosin according to standard protocol.

## 3. Results

### 3.1. Clinical Signs and Post Mortem (PM) Examination

Five juvenile suckling Egyptian fruit bats (*Rousettus aegyptiacus*) were collected from the ground (indicating that they were sick) and were admitted to the Wildlife medical center at the Safari Zoo Ramat Gan. Initial examination of bat 1334 showed pox like ring-type lesions on the ventral wings skin and abdomen from which skin swab was collected. The lesions included few superficial pale round plaques (Figure 1A, white arrows). The plaques were flat or slightly raised and consisted of mostly solitary 0.5 cm round or elliptic shape, which sometimes coalesced. The early lesions were pale in color and occasionally had a dark gray and erythematous border. The bats received antimicrobial drugs, nutrition and fluid support. However, no clinical improvement was observed and more pox like lesions appeared on the skin wing and advanced stages of lesion was notified (Figure 1B). Most of the lesions had depressed black punctual form centers with a pattern of varying design. Few days later, the bats died and the carcasses were sent to the Kimron Veterinary Institute. Bats 1374 and 1375 which were in contact with bat 1334 developed the pox like lesions a few days later. Four out of five bats (1334, 1374, 1375 and 1827) showed clinical signs for 6–8 days before death (Table 1). It is not clear if bat 1334 transmitted the virus to bats 1374 and 1375 or they were previously infected. Furthermore, in PM examination, four out five bats (1334, 1375, 1827 and 1840) showed: necrotic lesions, ulceration and perforation of the skin of the ventral wing close to the abdomen, as seen for bat 1334 (Figure 1C). In addition, few plaques were found on the abdomen. The pustules ruptured and crust forms were found on the surface of the ventral wing. This crust had sometimes become very thick. Some lesions healed often leaving a residual scar (Figure 1C). Bat 1374 showed only one plaque on the abdomen skin that developed later to a subcutaneous hematoma. At the necropsy, lesions were found only on the tongue of two bats (1374 and 1375). The lesion on the squamous epithelium of the tongue was gray round and 0.3–0.5 cm in diameter, with plaques and ulceration observed (Figure 2A, white arrow).

### 3.2. Histopathology

Pox like lesions were found on the wings skin (Figure 1D,E) and on the tongue (Figure 2B,C). However, no lesions were found in internal organs. On the wings skin, pox like lesions begin as cytoplasmic swelling, vacuolation and rounded degeneration of kerationocyte, with primary affecting the cells of the outer stratum spinosum (Figure 1D,E). Moreover, areas of spongiosis were seen as well (Figure 1E, black rectangle). Rupture of kerationocytes produces multiloculated vesicles, known as reticular degeneration. Examination of tongue tissues show abnormal keratinization with rounded degeneration and early dermal lesions. These lesions display congestion, edema, vascular dilation, a perivascular mononuclear cell infiltrate (Figure 2B,C). Neutrophils migrated into the epidermis and aggregated in vesicles to form microabscesses (Figure 2C). Furthermore, large intradermal pustules were observed which sometimes extended into the superficial dermis. A marked epidermal hyperplasia with focal ulceration, which, contributed to the raised border of the umbilicated pustules. Epithelial poxvirus lesions on the tongue showed characteristic of intracytoplasmic eosinophilic inclusion bodies of varying size appearing as single or multiple foci (Figure 2C). Interestingly, no poxvirus like intracytoplasmic eosinophilic inclusion bodies were found within the wings skin lesion (Figure 1D,E).

### 3.3. Molecular Diagnosis of Isolated Viruses by PCR

For initial molecular characterization of the isolated viruses, DNA was extracted from bats homogenized skin specimens. Using a published pan-chordpoxviruses PCR assay, we amplified a 220 bp and a 338 bp fragment of the viral Insulin metalloproteinase and DNA dependent DNA polymerase genes, respectively [17,18]. PCR amplification resulted in corresponding amplicons in diseased bats supporting the presence of poxvirus and its association with the observed clinical signs (Figure 3A,B). Furthermore, sequences analysis of these amplicons showed 100% nucleotide identity with the corresponding sequences of IsrRAPXV isolate 1000/15 recently published by us [15].

### 3.4. Virus Isolation

Vero cells were used to culture virus from bats skin swab. Four days post infection of Vero cells, a cytopathic effect (CPE) was visible on three samples (1334, 1374 and 1375), indicates by rounding up and locally detachment of cells from plate surface [15]. Subsequently, Vero cells were inoculated with homogenized skin tissues from bats 1827 and 1840 (with skin lesions) and a tongue lesion from bat 1375, all showing detectable CPE four days post infection. 10 days post infection, tissue culture plate was store at −80 °C and used for second passage and for future DNA purification and Molecular analysis.

### 3.5. Molecular Detection of IsrRAPXV by Novel qPCR

The observed CPE in Vero cells and the initial PCR diagnosis, both, support the presence of poxvirus in diseased bats, in particular, the association of IsrRPAXV in infected tissues. To test this, we designed a novel and specific real time qPCR assay for the detection of IsrRAPXV (Figure 4A). This assay allow us to efficiently and rapidly screen for the presence of IsrRAPXV in infected tissues. Additionally, it enable us to evaluate viral copy number and to correlate this with clinical signs. We first test our novel assay efficiency by using serial tenfold dilutions of a plasmid containing the qPCR target sequence (Figure 4B). The efficacy of qPCR was 100% with an R^2^ value of 0.995 and slope of –3.322. The limit of detection was 1 to 10 copies per reaction (Figure 4B). To further test assay specificity, we used a template from other pox viruses including Orf virus, Lumpy Skin Disease virus, Sheep pox virus, Camel pox virus and viruses from other families: Ovine Herpes Virus-2, Bovine Herpes Virus -1, Bovine Leucosis Virus and Bovine Adenovirus -3. None of these viruses yielded an amplification product using the IsrRAPXV specific qPCR assay (Figure 4C).

The new specific quantitative PCR was subsequently applied and successfully verified on DNA extracted from skin swabs and infected cultured cells (Figure 4D). Finally, we performed a rapid screen in order to teste viral load in different tissues extracted from diseased bats (Figure 5). Based on the screen we found that infected skin and tongue yielded the highest pox like viral DNA loads, while lower loads were found in inner organs (Figure 5). These findings are in agreement with our pathological findings, which show specific viral tropism for bats skin and tongue.

## 4. Discussion

Although bats serve as reservoir hosts for plethora of viruses, little evidence exists for associated death or illness of bats from viruses other than lyssaviruses [6]. This study provides evidence that link the recently discovered pox like IsrRAPXV [15] as the causative agent of morbidity, and might the cause of death, of juvenile suckling Egyptian fruit bats in Israel. In this study, we described the progression of clinical pox like disease in juvenile suckling Egyptian fruit bats (Figure 1 and Figure 2). In addition, we verified the presence of the poxvirus like IsrRAPXV, in dead bats, by molecular diagnosis and virus isolation in tissue culture (Figure 3). Moreover, we developed a novel qPCR assay for the rapid and efficient detection of poxvirus like IsrRAPXV in different bat organs.

Histological analysis of the skin wings of bats 1375, 1334, 1827 and 1840 is in agreement with the skin lesions we described before for IsrRAPXV [15]. However, no intracytoplasmatic inclusion were detected in present cases of skin lesions, which support our previous diagnosis (Figure 1, [15]). Nonetheless, histological analysis of the tongue epithelial lesions of bat 1375 showed the presences of eosinophilia intracytoplasmatic inclusion bodies (Figure 2). The pox like virus caused damage to the skin and tongue producing rounded degeneration of keratinocytes and epithelial necrosis followed by central ulceration (Figure 1 and Figure 2). The skin and tongue appear as highly productive sites for viral replication showing gross and histological lesions (Figure 1 and Figure 2).

Virus was successfully isolated either from swabs suspension (bats 1334, 1374 and 1375) or homogenized skin lesions (bats 1827 and 1840). Bats showed clinical signs for 6–8 days and eventually died, which imply severe infection (Table 2). It is important to mention that bats 1334, 1374 and 1375 were collected from the same geographical region and kept together until death. Therefore, it is not clear if bat 1334 transmitted the virus to the other two bats or they were infected prior to their cohabitation.

Our novel qPCR assay enable us to rapidly screen for viral presence in bat organs (Figure 4 and Figure 5). qPCR results indicate the presence of viral DNA in all tissue tested, suggesting that the pox like virus undergoes systemic spread within the host (Figure 5). Interestingly, variation of infected tissue were found between tested bats, although skin and tongue showed the highest viral load, supporting their role as primary site for viral replication and tissue tropism for IsrRAPXV (Figure 5).

In conclusion, our study demonstrates for the first time the development of sever disease and death in suckling bats, which involved poxvirus like lesions on the ventral wing skin and on the tongue.

## Figures and Tables

**Figure 1 viruses-13-00407-f001:**
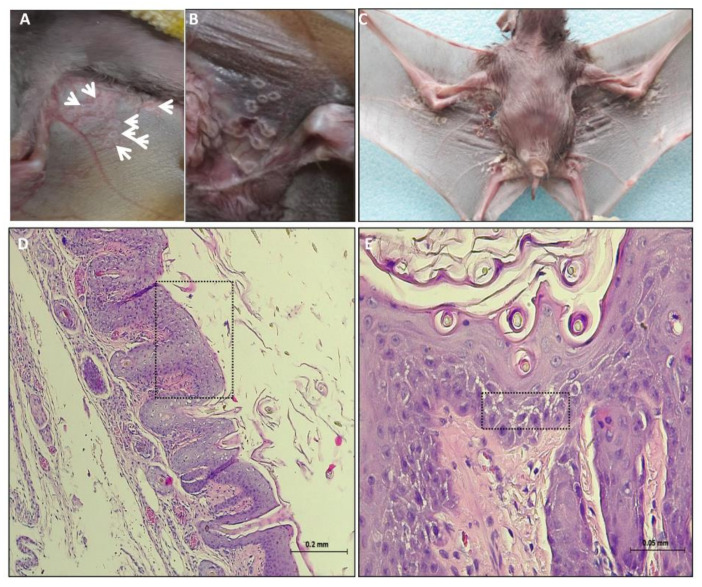
Development of skin lesions caused by poxvirus disease like in suckling bats. (**A**) Lesions on the ventral wing skin of bat no. 1334 marked by white arrows. (**B**) Progression of the Scheme in Figure 4. (**C**) PM examination showing gross pathology. (**D**) Histology of skin wing lesion, epidermis is hyperplastic, with pegs and nests extending into the inflamed dermis. Black rectangle point to a region with Acanthosis, vacuolar changes (ballooning) and reticular degeneration. (**E**) Higher magnification of (D) showing a dermal nest of epithelial cells with cytoplasmic rounded degeneration. Black rectangle point out a region with spongiosis.

**Figure 2 viruses-13-00407-f002:**
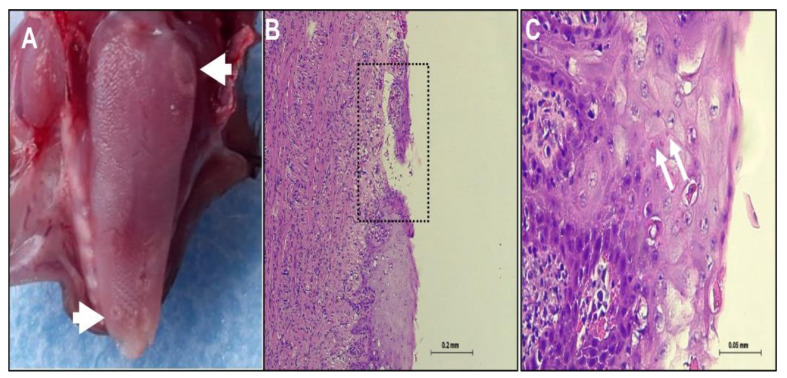
Development of tongue lesions caused by poxvirus disease like in suckling bats. (**A**) Plaques lesions on the tongue surface of Suckling bat no. 1375, mark by white arrows (**B**) Histology of tongue epithelia, showing ulceration (black rectangle, Magnification = 100×). (**C**) Intracytoplasmic eosinophilic inclusion bodies in the epithelial cells of the tongue (white arrows, Magnfication = 400×).

**Figure 3 viruses-13-00407-f003:**
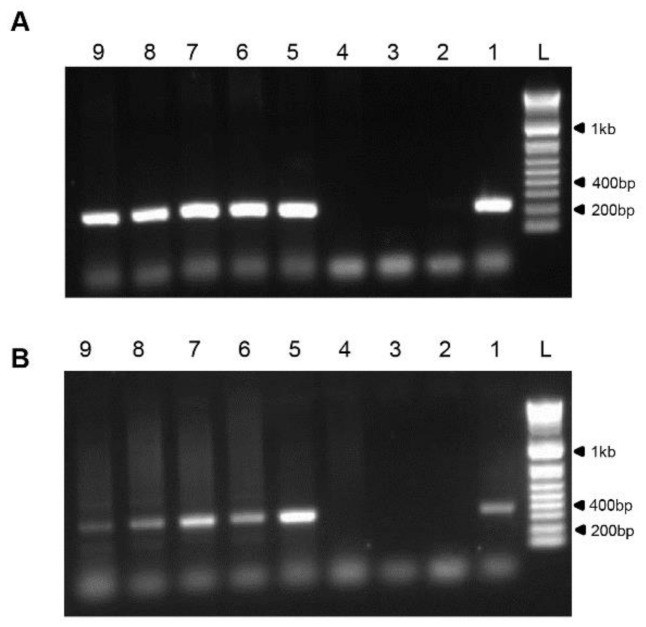
Detection of poxvirus in suckling bats. (**A**) PCR of poxvirus Insulin metalloproteinase-like-IMV region, amplifying 220 bp product. (**B**) PCR of poxvirus DNA-dependent DNA polymerase, amplifying 338 bp product. L- ladder (1 Kb plus DNA ladder, Invitrogen), 1-C+ from IsrRAPXV infected bat (strain 1000/15 (15)), 2- C- PCR mix, 3-C- isolation, 4-C- skin from healthy bat, 5-#1840, 6-#1827, 7-#1375, 8-#1374, 9-#1334.

**Figure 4 viruses-13-00407-f004:**
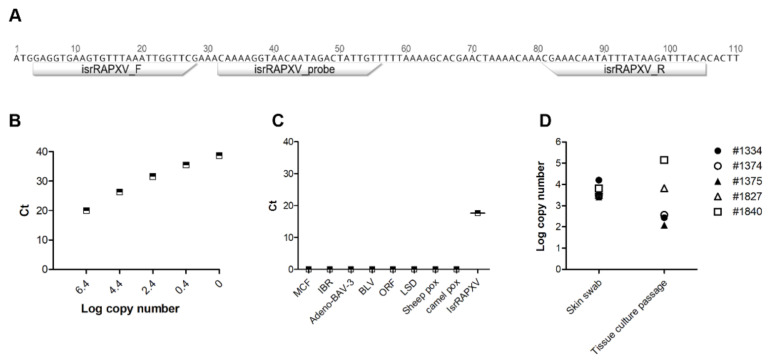
Development of RT-qPCR for detection of poxvirus in suckling bats. (**A**) Scheme showing partial sequence of E9L gene (Acc. No. MK542650) used to design isrRAPXV molecular detection assay. Annotations represent primers and probes location. (**B**) Standard curve with newly designed primer and probe for poxvirus molecular detection. (**C**) Specificity of the qPCR in detecting IsrRAPXV. (**D**) Detection of poxvirus in clinical samples and virus isolation on Vero cell line.

**Figure 5 viruses-13-00407-f005:**
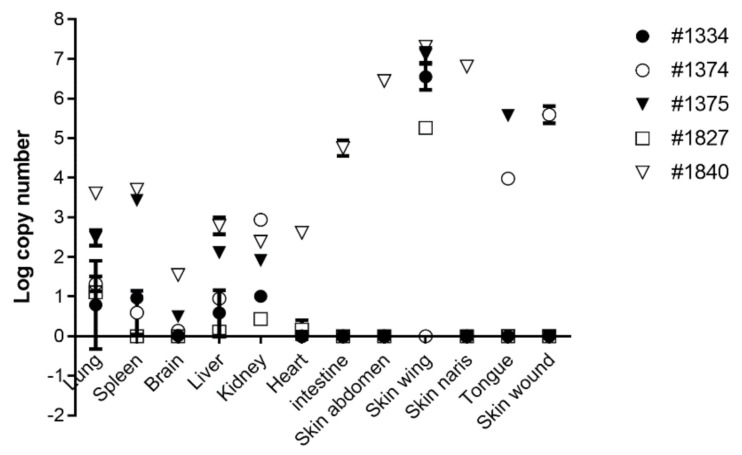
Detection of poxvirus nucleic acid in bats tissues. Molecular Detection of poxvirus copy number in tissues of diseased suckling bats by qPCR.

**Table 1 viruses-13-00407-t001:** Description and characteristics of Bats used in this study.

Bat (#)	Gender	Sample Location	Clinical Signs Detection (Date)	Clinical Signs Duration (Days)	Death	Sampling Date (Skin Swab)
1334	F	Qiryat Yam	18/3/19	6	23/3/19	18/3/19
1374	F	Qiryat Yam	23/3/19	8	30/3/19	23/3/19
1375	F	Qiryat Yam	23/3/19	8	31/3/19	2/4/19
1827	M	Ramat Gan	2/8/19	6	7/8/19	15/8/19
1840	M	Tel Aviv	NT	NT	20/8/19	25/8/19

**Table 2 viruses-13-00407-t002:** Primers and probe used in this study.

Name	Sequence of Primers (5′–3′)	Size (bp)	Reference
Poxpol_556F	GAYTAYAAYWSNYTNTAYCCNAAYGTITG	338	[17]
Poxpol_673R	RAANCCCATNARNCCRTAIAC		
Insulin metalloproteinase-like	ACACCAAAAACTCATATAACTTCT	220	[18]
IMV	CCTATTTTACTCCTTAGTAAATGAT		
IsrRAPXV-F	GAGGTGAAGTGTTTAAATTGGTTCG	102	This study
IsrRAPXV-Probe	FAM-CAAAAGGTAACAATAGACTATTGTT-BHQ1		
IsrRAPXV-R	TGTAAATCTTATAAATATTGTTTCG		
IsrRAPXV-037F	ATGGAGGTGAAGTGTTTAAATT	432	This study
IsrRAPXV-037411R	TCAGTGCAATGATAACACCCT		

## Data Availability

All data and results are enclosed in article.

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
