# Peer review of "Israeli Rousettus aegyptiacus Pox Virus (IsrRAPXV) Infection in Juvenile Egyptian Fruit Bat (Rousettus aegyptiacus): Clinical Findings and Molecular Detection"

_viruses, 2021, doi:10.3390/v13030407_

Round 1

Reviewer 1 Report

This article presents a clinical study of 5 juveniles of the species Rousettus aegyptiacus presenting initial local symptoms on the abdomen and the ventral wings skin, then a deterioration of the general condition with fatal outcome. Based on the clinical signs and a previously case identified in 2014 in an adult female of the same species (skin lesions without impairment of the general condition, which enabled the characterization of a new poxvirus called Israeli Rousettus Aegyptiacus Pox Virus (IsrRAPXV)), the authors suggested a poxviral infection. The cases described in the article confirm the presence of this virus in this population, and its ability to induce severe disease. In addition, the authors were able to develop new diagnostic tools (specific qPCR).

The article is interesting, allowing further characterization of a novel poxviral infection in a bat population (direct follow-up to the article by David et al, Vet Med Sci, 2020) and the development of a screening technique. Some points are nevertheless to be explained or developed:

Histology: It would be necessary in Figures 1d-e and 2c to show more specifically on the photos the essential histological lesions identified

Virus isolation: CPE was obtained after 4 days in Vero cells. What was the intensity of this 4-day CPE? Why did the authors freeze the infected cells 4 days after inoculation? Were there any other vero cell passages afterwards (to confirm the isolation)?

DNA extraction and qPCR:

How much tissue or organ was suspended in the 2 ml of pBS? Have the DNA concentrations after extraction been measured? Indeed, to be able to compare viral loads by qPCR, it would be necessary to know at least the total DNA concentrations of each sample and to perform the tests on equivalent quantities of DNA.

Why has the E9L gene of strain 1000/15 been amplified by RT-PCR (see line 100, page 3) and not by PCR directly (construction of the positive control plasmid)?

It would be useful to indicate the positions of the primers and probe on IsrRAPXV E9L sequence, and to present an alignment with E9L sequences representative of various poxvirus genus.

The results of the qPCRs are expressed in CT (Figures 4 and 5). However, a standard range was obtained (control plasmid used to measure the efficiency of the reaction and the limit of detection in genome copies/reaction), it is therefore possible to measure the viral loads in genome copies per reaction (or per microgram of DNA extracted). This information would be more relevant for the study of the different tissues and organs analyzed

During the qPCR specificity tests, only three ruminant poxviruses were tested (ORF, LSDV, SPV): why specifically one parapoxvirus and two capripoxviruses? why not have tested representatives of other poxvirus genus? The IsrRAPXV virus being very close to the Pteropox virus and the Sea Otter Pox virus, it could also be interesting to test DNA from these two viruses. Finally, it might also be useful to include DNAs from other poxviruses that have been identified on various bat species.

Illustrations: beware, figures 1D, 1E, 4 and 5 are too small

Author Response

Dear Reviewer #1

Thank you for your comments.

Please see below our point-by-point response:

Histology: It would be necessary in Figures 1d-e and 2c to show more specifically on the photos the essential histological lesions identified

A: We add labels to corresponding histological panels to point out histological changes. Please see the revised figures.

Virus isolation: CPE was obtained after 4 days in Vero cells. What was the intensity of this 4-day CPE? Why did the authors freeze the infected cells 4 days after inoculation? Were there any other vero cell passages afterwards (to confirm the isolation)?

A: We sorry for this confusion. On day 4 we observed CPE that characterized by rounding up of cells and plaques on multiple regions that eventually caused a cell detachment. Cells were collected 10 days post-infection and we did perform a second passage- this section was edited and revised to make it more clear to the reader (lines202-209).

DNA extraction and qPCR:

How much tissue or organ was suspended in the 2 ml of pBS? Have the DNA concentrations after extraction been measured? Indeed, to be able to compare viral loads by qPCR, it would be necessary to know at least the total DNA concentrations of each sample and to perform the tests on equivalent quantities of DNA.

A: We thank reviewer # 1 for this comment, we used ~10mg of tissues. For DNA extraction from bat organs, we did measure DNA after extraction and used 5ul per reaction in the original assay. According to Reviewer's comment, we performed this experiment again and used equal amounts of DNA to compare viral load in tissues – Please see revised Figure 5.

Why has the E9L gene of strain 1000/15 been amplified by RT-PCR (see line 100, page 3) and not by PCR directly (construction of the positive control plasmid)?

A: We sorry for this typo, word was edited to PCR.

It would be useful to indicate the positions of the primers and probe on IsrRAPXV E9L sequence, and to present an alignment with E9L sequences representative of various poxvirus genus.

A: We thank reviewer # 1 for this comment, we added a scheme showing primers and probe alignment to E9L gene, please see figure 4A. Because our qPCR is specific for the detection of isrRAPXV alignment with other poxviruses will end with no results.

The results of the qPCRs are expressed in CT (Figures 4 and 5). However, a standard range was obtained (control plasmid used to measure the efficiency of the reaction and the limit of detection in genome copies/reaction), it is, therefore, possible to measure the viral loads in genome copies per reaction (or per microgram of DNA extracted). This information would be more relevant for the study of the different tissues and organs analyzed.

A: Figures 4 and 5 were modified according to reviewer comment.

During the qPCR specificity tests, only three ruminant poxviruses were tested (ORF, LSDV, SPV): why specifically one parapoxvirus and two capripoxviruses? why not have tested representatives of other poxvirus genus? The IsrRAPXV virus being very close to the Pteropox virus and the Sea Otter Pox virus, it could also be interesting to test DNA from these two viruses. Finally, it might also be useful to include DNAs from other poxviruses that have been identified on various bat species.

A: Based on reviewer # 1 comment, we added one orthopoxvirus – camel pox to our figure. Please see revised figure 4. We tested our qPCR assay on E9L from both Sea otter pox and pteropox virus (in silico) and found that our assay is specific to isrRAPXV with no results on these two sequences.

Illustrations: beware, figures 1D, 1E, 4 and 5 are too small.

We thank reviewer # 1 for this comment, we modified these figures' size.

Reviewer 2 Report

The manuscript entitled "Israeli Rousettus Aegyptiacus Pox Virus (IsrRAPXV) infection in Juvenile Egyptian fruit bat (Rousettus aegyptiacus): Clinical findings and molecular detection" is well-written and the findings are scientificaly sound.  In light of the current SARS-CoV-2, virus detection in mammals, especially bats, is a very important area of research.  The paper is organized well and the methodologies are very straightforward and easy to follow.  I have only minor critiques of the manuscript:  

Line 29: Hyphen for SARS-CoV-2

Line 104: Escherichia coli should be italicized

Line 105: the “and” between ampicillin and Plasmid seems to be out of place

Figure 1:  The scale line is very difficult to see and the font is hard to read.  Please fix.

Figures 1 and 2: A line scale would be useful in images showing the wings and the tongue

Figure 3:  Please indicated the sizes of the bands in the ladder.

Lines 194-200:  The authors utilized Vero cells to isolate the virus and indicated CPE.  This data is indicated as “data not shown.”  Allowing the reader to see these images would strengthen the manuscript.

Line 209:  “Slope” instead of “slop”

Figure 4: Consider making this figure bigger to make it easier on the reader’s eyes

Figures 4C and 5: The author’s use Ct numbers to show the load is higher in the tongue and skin than other organs.  While the data is clear and gets the point across, the authors should consider showing the copy number in addition to the Ct as their novel assay gives them the ability to calculate

Author Response

Dear Reviewer # 2,

Thank you for your comments.

Please see below our point-by-point response:

 Line 29: Hyphen for SARS-CoV-2

A: Hyphen added as requested by reviewer # 2.

Line 104: Escherichia coli should be italicized

A: Font style change to italic as requested by reviewer # 2

Line 105: the “and” between ampicillin and Plasmid seems to be out of place

A:”and” was deleted from the sentence.

Figure 1:  The scale line is very difficult to see and the font is hard to read.  Please fix.

A: We thank reviewer #2 for this comment. We changed  Figure 1 order, enlarged the images, and upload a revised version. Scales are clearer now and were checked by printing the actual page.

Figures 1 and 2: A line scale would be useful in images showing the wings and the tongue

A: We thank reviewer #2 for this comment. Unfortunately, we cannot provide scale lines for wings and tongue images due to the fact both were taken with a hand camera during gross pathology examination of diseased bats.

Figure 3:  Please indicated the sizes of the bands in the ladder.

A: We thank reviewer #2 for this comment. We added bands size corresponding to our PCR product and upload the revised Figure 3. In addition, we added ladder information in figure legend.

Lines 194-200:  The authors utilized Vero cells to isolate the virus and indicated CPE.  This data is indicated as “data not shown.”  Allowing the reader to see these images would strengthen the manuscript.

A: We thank reviewer #2 for this comment. We agree that CPE images will support the data presented in the manuscript. However, we already published CPE in VERO cell infected with isrRAPXV and in order to avoid repetition of data we wrote data not shown. We will delete (“data not shown”) and cite our previous publication instead.

Line 209:  “Slope” instead of “slop”

A: We fixed Typo.

Figure 4: Consider making this figure bigger to make it easier on the reader’s eyes.

A: We enlarged Fig. 4 and uploaded a revised figure.

Figures 4C and 5: The author’s use Ct numbers to show the load is higher in the tongue and skin than other organs.  While the data is clear and gets the point across, the authors should consider showing the copy number in addition to the Ct as their novel assay gives them the ability to calculate.

A: We thank reviewer #2 for this comment, we uploaded a revised Figures 4 and 5, both showing copy numbers.

Reviewer 3 Report

In this brief report/study the authors describe a novel qPCR assay to detect the presence of Israeli Rousettus Aegyptiacus Pox Virus in the lesions and organs of Egyptian fruit bats. In experiments following on from their previous publication (David et al, 2020) they have further identified Rousettus Aegyptiacus Pox Virus in five further juvenile suckling Egyptian fruit bats and analysed the clinical and pathological findings.

I found the paper very easy to read and the methodology easy to follow. Occasional minor spelling and/or language edits are required.

Minor alterations:

Line 29: Would be better to use the source ref for this: Boni et al.

Line 56: contaminant should be replaced with concomitant

Line 109: 5L of extracted DNA?

Line 167: how should read show?

Author Response

Dear Reviewer # 3,

Thank you for your comments.

Please see below our point by point response:

Minor alterations:

Line 29: Would be better to use the source ref for this: Boni et al.

A: ref changed as suggested by reviewer # 3.

Line 56: contaminant should be replaced with concomitant

A: We fixed the typo and replaced the word as suggested by reviewer # 3.

Line 109: 5L of extracted DNA?

A: We thank reviewer #3 for pointing out this typo- we fixed it in the manuscript. 

Line 167: how should read show?

A: We thank reviewer #3 for pointing out this typo- we add S to the word and now the sentence is clear to the reader (line 170 in the revised manuscript).